# Tourism Sustainability Index: Measuring Tourism Sustainability Based on the ETIS Toolkit, by Exploring Tourist Satisfaction via Sentiment Analysis

**Damiano De Marchi *** , **Rudy Becarelli and Leonardo Di Sarli**

The Data Appeal Company, Via del Tiratoio, 1, 50124 Firenze, Italy; rudy.becarelli@datappeal.io (R.B.); leonardo.disarli@datappeal.io (L.D.S.)
* Correspondence: damiano.demarchi@datappeal.io; Tel.: +39-345-1563011

**Abstract:** Interest in measuring the sustainability of tourism has been significantly advancing in recent years, together with the need to manage the impact of tourism on territories and hosting communities. This interest was further boosted by the pandemic, with sustainability being identified as one of the central elements in restarting global tourism. The European Tourism Indicator System (ETIS), developed by the European Commission, is a point of reference based on self-assessment, data collection, and analysis by tourist destinations themselves. The application of the ETIS toolkit has faced many challenges, especially at the subnational level, most of which are related to the lack of available and updated data to feed into the model. In this article, we explore the implementation issues, develop a synthetic indicator based on the use of the sentiment analysis technique to frame e-reputation and tourism satisfaction, and combine that analysis with other open data sources. The Tourism Sustainability Index (TSI) can provide a scalable and georeferenced evaluation of tourism sustainability, measured according the ETIS criteria and complementing them. The TSI, its pillars and sub-components are all applicable to any tourism destination. The results show that the TSI can be a consistent and valid tool for tourist destinations to use in analyzing sustainability, monitoring the evolution of sustainability through time periods and subareas, and comparing the findings with those of other benchmarks and/or other competitive areas.

**Keywords:** tourism; measuring sustainability; tourist satisfaction; e-reputation; sustainable development; sentiment analysis; ETIS; open data; geospatial index



## 1. Introduction

### 1.1. Measuring Tourism Sustainability

Interest in sustainable tourism has been advancing since the early 1990s, with the United Nations World Tourism Organization (UNWTO) promoting the use of sustainable tourism indicators as essential instruments for policy making, planning, and management processes at destinations [1]. In the new millennium, sustainable tourism assessment became increasingly relevant in national and international agendas, culminating in the UN's declaration that 2017 was the International Year of Sustainable Tourism for Development, following the adoption of the United Nations 2030 Agenda for Sustainable Development and the Sustainable Development Goals (SDGs). The recognition of tourism's potential to contribute, directly or indirectly, to all of the sustainable development goals (SDGs) boosted the need for an initiative to develop a common framework for measuring the environmental, economic, and socio-cultural impacts of tourism at every level. The Statistical Framework for Measuring the Sustainability of Tourism, launched by the UNWTO with the support of the United Nations Statistics Division, aimed to develop an international statistical framework for measuring tourism's role in sustainable development, including economic, environmental, and social dimensions, and to support universal, cross-sectorial,

and sustainable tourism policies and practices that operate from an integrated, coherent, and robust information base.

Although scientific literature and practical applications have underscored the difficulties of applying the principles of sustainable development to a specific sector, the need for a valid measurement for tourism sustainability is evident across all stakeholder groups, cutting across global, national, and subnational (including local) levels, and a key tool to implement the principles of sustainability in the tourism industry is required [2,3].

It has been acknowledged that defining a set of indicators that are internationally comparable to enable assessments of sustainability and the improvements that are needed at tourist destinations will allow tourism managers to monitor the situation and, eventually, identify the actions that are required to improve the level of sustainability across the different elements of tourism [4]. In 2016, the European Commission developed a "European Tourism Indicators System" (ETIS), aiming to provide all of the specific tools to monitor the impact of tourism at the local level. The ETIS monitoring system helps tourist destinations to measure their performances with respect to sustainability by using a common and comparable approach. The ETIS model is composed of four pillars and 43 core indicators, together with an indicative set of supplementary indicators. Its monitoring results are based on self-assessment, observations, data collection, and analysis by the destinations themselves [5]. The flexibility provided by the system—destinations can choose for themselves the most relevant indicators they wish to adopt—was initially designed to improve the ETIS's feasibility and potential for success. However, the task of tailoring the indicators to a particular destination sometimes became an obstacle to actual implementation [6]. Moreover, the lack of data needed to implement the system, together with tepid interest among destination stakeholders in adopting the ETIS as a long-term investment (to cover, for example, the costs of conducting surveys on visitors, residents, and tourism businesses), presented problems for tourism destinations [7]. The recorded experiences of implementing the ETIS within Italian tourism destinations revealed several critical issues, indicating that new technologies are needed to facilitate the collection of statistical data [8,9].

Despite these challenges, during two pilot phases in 2013 and 2015, the ETIS was voluntarily implemented by more than 100 destinations; it remains the best-known and most-cited methodology for measuring tourism sustainability at a destination level.

### 1.2. Tourism Sustainability and Reputation

The concept of sustainability has been widely accepted as a means of mitigating the impacts of tourism. Since the early 1990s, it has been linked intensively to the carrying capacity of tourism destinations. The critical range of elements of this carrying capacity includes consideration of unsustainability, harm to a destination's elements of attraction [10], the quality of visitors' experiences, and the negative effects of tourism on third parties that are not directly involved in tourism activities [11,12]. In recent times, customer satisfaction has been added as another key element to be considered when discussing tourism's carrying capacity [13], as it may be directly related to the other elements of sustainability, as perceived by different users within a tourist destination, i.e., residents (whether or not they work in the tourism industry), commuters, visitors, tourists, and excursionists. The ETIS model includes customer satisfaction as one of the indicators to be measured.

Consumer perception became an important element for consideration as the concept of risk became more significant. In studies of consumer purchasing behavior, a great deal of attention has been devoted to the concept of risk, as the increasing average life span of consumers has led them to reflect increasingly on risk, which, in turn, leads societies to demand ever higher levels of safety [14].

The search for timely and precise information is a determining factor for individual and collective users in the process of mitigating the perception of risk, whether the risks are statistically and objectively demonstrable or based on perception (which is related to cognitive, emotional, and social dimensions) [15]. Over the years, most of the relevant

literature demonstrated the importance of emotion in risk perception and risk-taking behaviors and the fundamental role of emotional reactions in risk judgment and decision making [16,17]. Accordingly, reducing uncertainty plays a key role in the process that leads to a decision to make or not to make a purchase, especially in a virtual transaction, where a customer includes the financial risk of the transaction in assessing the potential technical complexity of the purchase. The development of the Web 2.0 in the early 2000s encouraged people to express their opinions about products and services through social media [18], and many studies focused on the correlation between user-generated content (UGC) and sales. Online reviews of products and services have a significant impact on both consumers' purchasing decisions and sales [19–22].

For this reason, as e-commerce began to soar, marketplaces such as Amazon, Ebay, and Booking.com included platforms for registered users to share opinions about their buying experiences. In reducing the risks for customers by providing information before actual purchasing decisions [23], and in attracting new customers by providing online platforms that enable them to exchange their consumption experiences [24], online companies developed more sophisticated methods to enrich the reviewing experience, by, e.g., including ratings, pictures, and reviewer perspectives.

In tourism, risk mitigation is particularly important, for a variety of reasons, including the characteristics of tourism products—their intangibility, the inseparability between provider and customer, the absence of tourist ownership, and consumer heterogeneity. The purchase of tourism products often occurs online; the costs of tourism include money, time, and emotional investment; the consequences of a bad tourism experience may be perceived as significant. In investigating the perceived risks of tourism, many elements have been identified [25].

It is clear that the reputations of tourism destinations have a direct influence on the purchase of tourism services as well as on how visitors approach and experience a destination and, consequently, how they develop a process of involvement and trust, which are keys to success for businesses of any size and location [26–28]. Hence, to understand the impact of reputation on visitors' opinions, the new research field of sentiment analysis emerged during the last decade. It is generally defined as a computational process of recognizing, detecting, and determining the orientations and polarities of human opinions or emotions [29,30], based primarily on big data and tourists' user-generated content (UGC) [31]. The use of sentiment analysis in tourism can contribute positively to the decision-making processes of governments and businesses in fostering sustainability and understanding consumer behavior patterns [32,33].

### 1.3. Towards a Tourism Sustainability Index (TSI)

"Sustainability must no longer be a niche part of tourism but must be the new norm for every part of our sector. This is one of the central elements of our Global Guidelines to Restart Tourism. It is in our hands to transform tourism and that emerging from COVID-19 becomes a turning point for sustainability."

With these words in 2020, UNWTO Secretary-General Zurab Pololikashvili addressed the One Planet Vision for the Responsible Recovery of the Tourism Sector, a repository of inspiring initiatives, tools, and strategic thinking, representing a common vision for better tourism for people, the planet, and prosperity [34]. Considering the fundamental importance of sustainability for tourism, it is indispensable that its principles be adopted by any type of destination to manage and mitigate the impacts of the tourism phenomenon. Moreover, sustainability must be measured, together with its elements, such as governance, environmental impacts, economic impacts, and social–cultural influences, to make it a strategic objective in management plans and actions.

The challenges to a widespread implementation of the ETIS methodology, which are emerging after the first two pilot phases, have yet to be solved by the existing tourism sustainability indices. The sustainable tourism index, created by the Economist Intelligence Unit [35], and the tourism sustainable development index, developed by the European

Space Agency in collaboration with the Murmuration organization [36], can be applied only at the country level. The global destination sustainability index, designed by the Global Destination Sustainability Movement, can be applied at the destination level, but it is based on a questionnaire that requires the collection and submission of quantitative and qualitative data by the destinations themselves [37]. The ISOST index (a tool for studying tourism sustainability) [38] has remained at test level and has yet to be continuously implemented by tourist destinations. Therefore, the implementation of a genuine and widespread measurement of tourism sustainability requires the combination of a survey-based approach with data sources that are self-updating.

The main contributions of this paper are presenting an index to measure tourism sustainability based on the ETIS toolkit and exploring tourist satisfaction via sentiment analysis combined with open data sources. The research hypothesis was that sentiment analysis can supplement surveys by providing an automatic and scientific process for measuring the satisfaction of different tourism users. Our approach takes UGCs into consideration and defines the relevant volumes, variations, and trends to establish the index.

We adopt a pragmatic and original approach in presenting our Tourism Sustainability Index (TSI), which is a georeferenced synthetic indicator that evaluates, via a quantitative scale (0–100), the sustainability of tourism activity in a destination. The TSI is scalable; it is suitable for any location worldwide; and it provides a monthly time frame based on various sources of data, as described in the following section, including sentiment analysis as the main approach to any of its four pillars, defined according to the ETIS methodology. We present the TSI to the scientific community in anticipation of test applications for tourism-related territories, consideration of managerial implications, and possible future research.

## 2. Methodology

The methodology presented in this paper describes the authors' approach in designing the Tourism Sustainability Index with the procedures and proprietary algorithms of The Data Appeal Company (TDAC)—Florence, Italy—based on a machine learning model that analyzes online conversations and provides a multi-class score for any text [39].

### 2.1. The Process

In order to develop the TSI, the process starts from the definition of points of interest (POIs), which are defined as economic, cultural, and naturalistic geographical points liked to a set of textual contents that can be retrieved and considered. TDAC collects and monitors the digital presence of POIs through an analysis of over 130 online platforms, websites, online travel agencies (OTAs), and social media channels (e.g., Google, Booking.com, Airbnb, TripAdvisor, Facebook, and Twitter) and explores the characteristics of POIs and UGCs. A complex automatic process aims to maximize the probability that the channels explored are related to the same POI, comparing and completing the information through the various channels to build the digital identity of the POI. Considering the amount of data involved, a crucial factor for the correct use of these data is the activation of data quality procedures to identify and possibly correct outliers, anomalies, and/or potential inconsistencies. In addition, the data quality process must be automatized, with fast and precise algorithms that are designed for the specific problems to be monitored [40]. Once the ingestion and data quality processes have been completed, the data are stored in TDAC's private data lake. At this time, semantic analysis takes place, processing the textual contents with the aims of providing a polarity score (sentiment) for each content and identifying the main topics and the opinions connected with these topics. The algorithm is based on three models:

- The name entity recognition (NER) model, which aims to classify words or phrases within predefined categories, starting from unstructured texts such as reviews;
- The dependency parser model, which analyzes the grammatical structure of texts and identifies the connections between the various words;

- The sentiment analysis model, which is a classic machine-learning natural language processing (NLP) model that looks for nonlinear dependencies between the various words to "understand", at a computational level, the logic that represents the satisfaction and the polarity of a generic text.

Sentiment analysis was the key model for the development of the TSI. As further detailed in the following section, the TSI was developed on the basis of two sources: public data (also known as open data) and proprietary data. Open data were used to assess the objective components of the index, while proprietary data were mainly used to assess the subjective components of the index. As result of sentiment analysis, proprietary data provided customers' perceptions of the experiences related to the tourism industry. Because sentiment analysis is applied to UGCs in relation to each POI, and these UGCs vary over time, it was possible to average the sentiment groupings of the POIs by type of sector, time, and geography (i.e., the average/median sentiment computed for hospitality-related POIs in a certain county during a certain time period).

The same approach could be used to deduce the sentiment related to a certain territory or to a particular event if sentiment analysis was applied to social media contents filtered by hashtag. The implementation of objective and subjective points of view provides an unprecedented insight into tourism sustainability, as the TSI considers not only the typical environmental parameters but also users' perceptions of policy efficiencies, with the aim of preserving socio-cultural heritage and local productions.

Finally, to ingest and aggregate publicly available data, three different activities were carried out:

- Investigating data sources that could be useful in characterizing all aspects of sustainability as defined in the ETIS methodology;
- Defining and implementing processes that must be put in place for data cleaning, data quality, and data understanding. This activity was necessary because heterogeneous data from previously selected data sources needed to be synchronized on the same timeline, and missing or spurious data needed to be reconsidered and/or geographically resampled in order to profitably aggregate a georeferenced indicator;
- Designing, creating, and aggregating each element of the final index. Each component was implemented as a specific model to describe each aspect of the final index, following an additive/multiplicative aggregation approach in which a compensation between the base indicators was admitted [41]. The list of all the components used in the final index computation, according to pillar and rationale, will be detailed in the last part of this section.

### 2.2. The Data Sources

A heterogeneous data set was selected to estimate the compliance of a tourist destination with the objectives of the TSI. These data had the following characteristics: they must be scalable (i.e., they must cover the entire planet even at the cost of undermining accuracy and resolution, depending on the country to which they are applicable); they must be georeferenced (i.e., they may be localized on a map with predicted precision); and they must be storified (i.e., they can be ingested according to a certain schedule as they change over time. Their frequency can be variable, depending on the nature of the content).

To cover a large area and to deal with data outliers, the index was computed with reference to tile coverage, as provided by the Bing Maps Tile System [42]. The size of the tile halved each time the level increased, starting with level 0, which covered the entire Mercator projection of the globe. The level of detail we chose, the Bing Tile level 16 with a tile size of 611.5 × 611.5 m, permitted the sustainability analysis to be deepened at the sub-destination level, which was a significant advantage for strategic and operative planning. The various data sources involved in the process had their own geographical and temporal resolution, so it was often necessary to geometrically interpolate, average, or impute any missing data to provide a full representation of the involved phenomena, at both the spatial and temporal scales. Furthermore, data sources were divided into two

different categories: open data sources and proprietary data sources. The following section details their characteristics and their ingestion technique.

### 2.2.1. Open Data

Open data refers to any data that are openly accessible, exploitable, editable, and shared by anyone for any purpose. In the remainder of this analysis, the term open data is used for the datasets provided by two different sources:

- Copernicus [43], the European Union's Earth observation program, which offers information services that make raw and preprocessed data available; the data are gathered by satellites that continuously observe and measure the planet's environmental parameters. Another set of relevant data comes from the direct (in situ) measurements that are accomplished by means of ground-based, airborne, and seaborne platforms. These additional data come with a higher degree of accuracy and precision, at the cost of less scalable coverage. The European Commission manages the program in partnership with the member states, the European Space Agency (ESA) [44], the European Organization for the Exploitation of Meteorological Satellites (EUMETSAT) [45], and the European Centre for Medium-Range Weather Forecasts (ECMWF) [46]. The information services are free and openly accessible to any registered user;
- The World Bank Data Portal [47], which provides access to global economic and development statistics, including World Development Indicators, International Debt Statistics, Millennium Development Indicators, and data on poverty, education, and gender-related issues (e.g., disparity in economical revenues and social/human rights related to gender). These kind of data were useful in defining the social and economic context for some components of the final index, namely, for components that measure the impact of tourism on local cultures and economies.

### 2.2.2. Proprietary Data

Proprietary data refers to data that are stored and managed in The Data Appeal Company's data lake. TDAC simultaneously combines three key, real-time intelligence elements: location, sentiment, and market intelligence. The data lake provides access to both historical and forecasting insights for any point of interest or territory. The Data Appeal Company is able to gather data from hundreds or thousands of UGCs each year, related to tourist attractions and facilities in definite locations. The amount of data depends on different factors, such as the location size, seasonality, and the digital presence of online channels for different POIs; the most recent figures indicate over one billion contents of data worldwide, on almost 10 million POIs. The considerable amount of UGCs and the sentiment analysis that can be derived from the data describe a trend that can be used to establish the TSI. In fact, with reference to the qualitative value of that data, several studies conducted during the past 2 years in collaboration with actual customers (i.e., the Italian government, the Veneto region, the Tuscany region, and other customers at subregional levels, described as provinces or municipalities) show the correlation between changes in online content and tourism arrivals, with a Pearson correlation coefficient that is always >0.90 [48]. Therefore, the use of such data as an indicator of tourism flows maintains the same information about trend patterns, easing or even eliminating the need to constantly have an updated source of official tourism-flow data for a specific destination. Proprietary data forms the backbone of all of the components of the index, while datasets coming from open data sources provide supportive, but still relevant, information that is useful in orchestrating and harmonizing all of the index's components within and among the tiles covering a particular destination.

### 2.3. The Components of the Tourism Sustainability Index

Consistent with the ETIS methodology, the TSI has four main components (pillars), each of which is composed of several sub-indicators to map the ETIS toolkit with sustainable destinations criteria. Our approach in composing the TSI was first to list all of the ETIS

criteria, then to select the ones that meet the requirements of scalability and availability of proprietary and open data. As stated, one of the main limitations for widespread implementation of the ETIS toolkit is that most indicators can be sourced only by conducting surveys, questionnaires, and other activities in the target area. For these indicators, there is not a usable open data lake, because not all tourism companies or institutions collect the necessary data, so they must be considered after direct interaction with tourists and tourism stakeholders, leading to the need for human supervision. Therefore, some ETIS criteria were discarded, including the following: the tourism supply chain, defined as the percentage of locally produced food, drinks, goods and services sourced by the destination's tourism enterprises (B.4); the solid waste management (D.3). Our selection of indicators that meet the requirements of scalability, availability, and scientific significance allowed us to compute the TSI for every single tile, after robust data collection and establishing data quality through an automated process. As detailed in Table 1, the remaining sub-indicators provided a good representation of the core dimensions and various aspects included in each of the ETIS pillars, with notable importance assigned to the topic of satisfaction explored via sentiment analysis. We consider the proposed TSI as a minimum viable product (MVP) for achieving testing results and the consequent validation process, as described in the following sections.

**Table 1.** Linkages between ETIS criteria and TSI sub-indicators, with details of the data sources used.

| ETIS Criteria Reference | TSI Sub-Indicators | Sentiment Analysis | Data Source |
|---|---|---|---|
| A.2 Customer satisfaction | Customer Satisfaction Index<br>Short-Term Confidence | Yes<br>Yes | Proprietary data<br>Proprietary data |
| B.1 Tourism flow at destination | Seasonality Balance<br>Volume of Visitors<br>Percentage of Business Tourism | Yes<br>No<br>No | Proprietary data<br>Proprietary data<br>Proprietary data |
| B.2 Tourism enterprise(s) performance | OTA Penetration | No | Proprietary data |
| C.1 Community/social impact | Activity of Short-Term Rentals<br>Tourism Pressure<br>Tourism Supply Pressure | Yes<br>No<br>No | Proprietary data<br>Proprietary and open data<br>Proprietary and open data |
| C.2 Health and safety | Good Health Index | Yes | Proprietary data |
| C.4 Inclusion/accessibility | Accessibility Index | No | Proprietary data |
| C.5 Protecting and enhancing cultural heritage, local identity, and assets | Cultural Index<br>Urban Green Index | Yes<br>Yes | Proprietary data<br>Proprietary data |
| D.1 Reducing transport impact | Public Transportation Accessibility | Yes | Proprietary data |
| D.2 Climate change | $CO_2$ Index<br>Air Quality Index<br>Sea Quality Index | No<br>No<br>No | Open data<br>Open data<br>Open data |
| D.7 Landscape and biodiversity protection | Natural Coverage | No | Open data |

All of the TSI sub-indicators were computed in a range between 0 and 100 in order to be descriptive on their own with respect to their specific domains. All of these sub-indicators were then combined in an analytical formula to build the four pillars and representing the overall TSI. This specific range allows users to compare different behaviors in an easy way, and facilitates the calculation of the TSI.

### 2.3.1. Destination Management Pillar

Because no data source with the above characteristics can provide the governance of a destination, the destination management pillar emphasizes the perception, confidence,

and success of a tourist destination. The data that were used to compute the component indicators were mainly proprietary data. In particular, sentiment analysis and the contents/time series analysis were used. All of the involved sub-indicators were subject to on sentiment analysis.

### 2.3.2. Overtourism Pillar

In the pilot stage, it was impossible to reach an economic indicator based on data sources with all of the above characteristics. Therefore, we proposed a pillar emphasizing overtourism in a destination, mixing different indicators regarding tourist flows, pressures on tourism supply, pressures on hosting communities in terms of satisfaction, and the percentage of accommodation that, according to our estimates, were booked via online travel agencies (OTAs). These indicators were computed with proprietary data, such as sentiment, a time series analysis of online content, and the point of interest composition of a destination, combined with World Bank Data, such as population density.

### 2.3.3. Social and Cultural Pillar

The social and cultural pillar focuses on the effects of tourism on both the social and cultural aspects of a destination, such as the development and efficiency of healthcare systems, the presence and quality of urban green, the destination's cultural vitality and reputation, the presence of events, and business meetings, incentives, conferences, and exhibitions (MICE). A special focus was given to accessibility, i.e., how many POIs are accessible to disabled tourists and general accessibility in terms of public transportation. The main data source used to compute the following indicators was proprietary data, particularly the sentiment analysis and the contents/time series analysis. Various sub-indicators were derived from sentiment analysis, as various topics related to urban green, cultural experiences, and good health evaluations were identified by UGCs.

### 2.3.4. Environment Pillar

The environment pillar focuses on elements that are critical to the sustainability of the environment of a destination, highlighting the importance of investing in environmental protection and assessing the impact of tourism on the environment. High quality of air and water and great biodiversity help ensure the sustainability of natural areas, benefit a destination's image, and attract visitors. This pillar highlights both the importance of investing in landscape and biodiversity protection and the tourism sector's role in supporting this process. This pillar and its indicators were computed on Copernicus open data. In this stage, the pillar was mostly related to objective data (land coverage, air/sea quality) and the sentiment analysis contribution was relevant only with regard to public transportation accessibility.

## 3. Results

In order to test the TSI, we deployed a front-end data visualization tool. in this way, it was possible to apply the TSI to a customer area and test the results. A double round of tests was carried out: first, a single area was tested to explore, compare, and inspect the TSI and its components; then, different areas were tested to investigate the potential uses and practical implications of the TSI. In both cases, a selection of snapshots was presented. The analysis, data collection, and data pre-processing steps were carried out in April 2022.

### 3.1. Testing the Index: Applying the TSI to the Province of Milan

There are several reasons for the Province of Milan to be designated. It is a wide area containing a metropolitan city (Milan), other towns, and small villages. The area presents a variety of characteristics: the presence of high-quality hospitals and health centers, public services, and facilities, but a severe concentration of air pollutants and $CO_2$, as well as intensive land use. Milan is a major tourism destination for domestic and international visitors, combining several different options, such as leisure and recreation, business and

professional purposes, visits to friends and relatives, healthcare, study opportunities, and events. More than 75,000 POIs were analyzed to test the TSI in the Province of Milan, applying semantic analysis to over 2.9 million contents, of which roughly 1.8 million were related to tourism industries for the whole of the year 2021.

The first test results are shown in Figure 1, presenting the value of the Tourism Sustainable Index and its four pillars for the Province of Milan. The TSI were computed as a weighted average of the four pillars: scores have a value in a range between 0 and 100. A higher score means a higher level of sustainability.

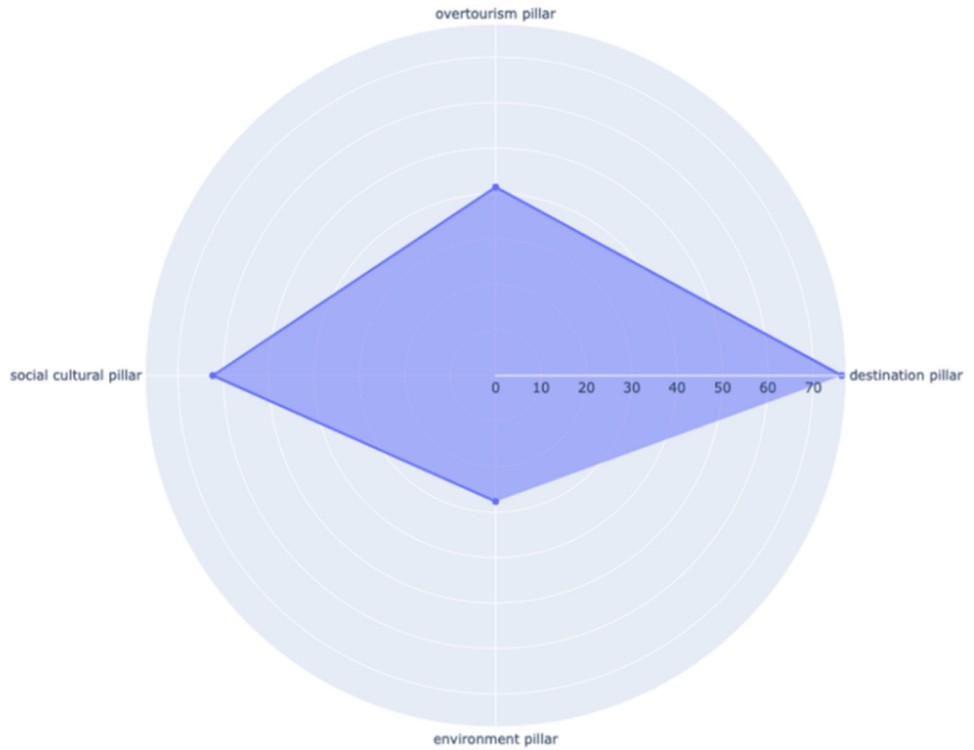

| Destination Pillar | Overtourism Pillar | Social Cultural Pillar | Environment Pillar | TSI |
|---|---|---|---|---|
| 76.29 | 41.42 | 62.35 | 27.69 | 47.07 |

**Figure 1.** TSI and its pillars, 2021: Province of Milan.

A selection of five snapshots from the front-end tool is presented to thoroughly examine the target area and explore the subcomponents of the TSI, focusing on the ones that are derived from UGCs and sentiment analysis. A high level of sustainability corresponds to a higher score or a darker tile.

In Figure 2, the customer satisfaction index is applied. It measures the tourist satisfaction via sentiment analysis of UGCs. Even if the whole province of Milan were to record a very good average score for a year (85.1/100), it is clear that the distribution of the score differs significantly. The areas with a higher satisfaction (the darker areas) are located on the boundaries of the county rather than in the center.

One of the reasons for higher levels of tourism satisfaction recorded in peripheral areas of the Province of Milan rather than in the center is the spread of the tourism pressure. High tourism pressure can significantly affect the tourist experience and lowering levels of satisfaction. In Figure 3, the tourism pressure indicator is applied. It measures the effect of tourism online contents, derived by UGCs analysis, on residents. The component shows how the center of Milan has higher tourism pressure compared to that of its peripheral area, which corresponds exactly with the official tourism data flows registered by the statistics authority.

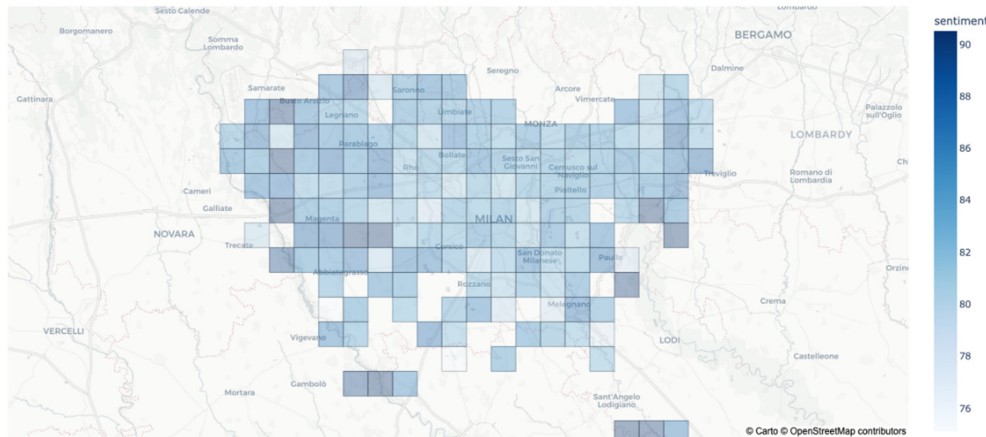

**Figure 2.** Customer Satisfaction Index (Destination Pillar), 2021: Province of Milan.

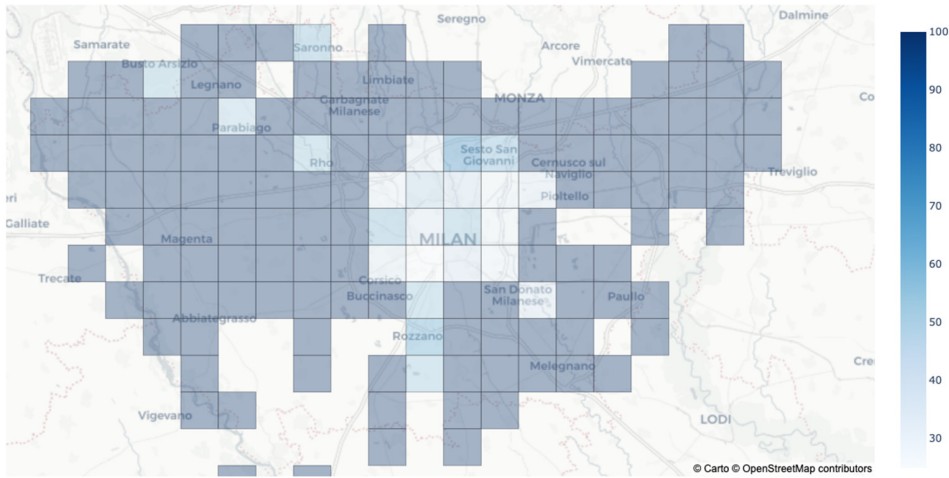

**Figure 3.** Tourism Pressure (Overtourism Pillar), 2021: Province of Milan.

In Figure 4, the good health index is applied. It measures the concentration, quality, and e-reputation of hospitals via sentiment analysis of UGCs that are related to healthcare centers. The best medical centers in this area are located in the two darkest tiles, which correspond with the areas of five hospitals that are listed in the World Best Hospitals 2022 Report [49].

To present the results of tests of environmental sub-indicators, we selected one indicator related to objective data and one indicator that included sentiment analysis. In Figure 5, the air quality index is applied. It measures concentration in terms of several air pollutants: $PM_{2.5}$, $PM_{10}$, $NO_2$, $SO_2$, and $O_3$. It was evident that the metropolitan area of Milan, as well as some other industrial areas, suffered from bad air quality, which corresponded with the results of a recent analysis by Legambiente, who reported that Milan was one of the most polluted cities in Italy [50].

In Figure 6, the public transportation accessibility index is applied: it measures the quantity, concentration, quality, and e-reputation of public transportation via sentiment analysis on UGCs that are related to public transportation. The results showed that the center of Milan presented the highest level of accessibility via public transportation in the whole province, with several different means of transportation, such as subways, buses, and trains. In the suburbs, the tiles are lighter, in particular in the western area, which is the part of the Province of Milan that is less densely served by the regional train network and suburban public transport services.

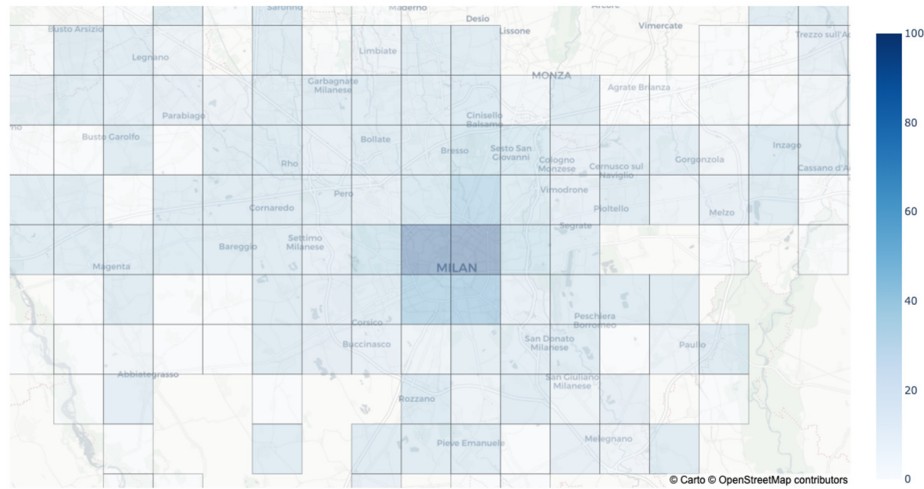

**Figure 4.** Good Health Index (Social-Cultural Pillar), 2021: Province of Milan.

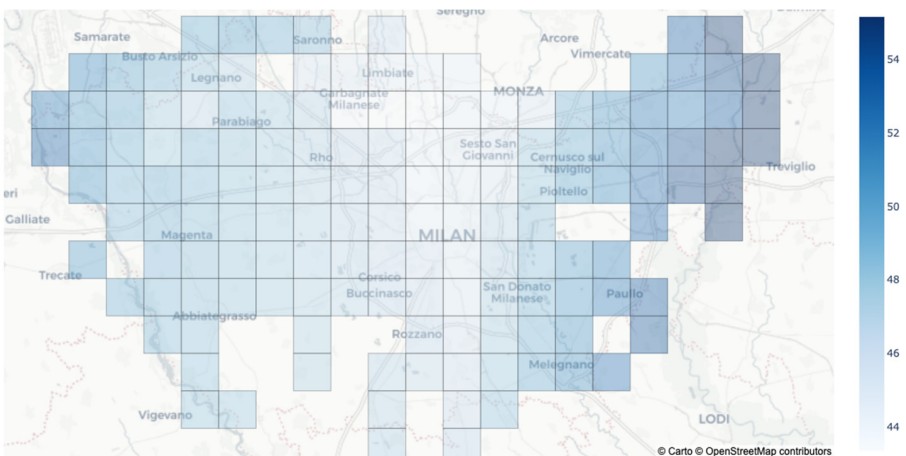

**Figure 5.** Air Quality Index (Environment Pillar), 2021: Province of Milan.

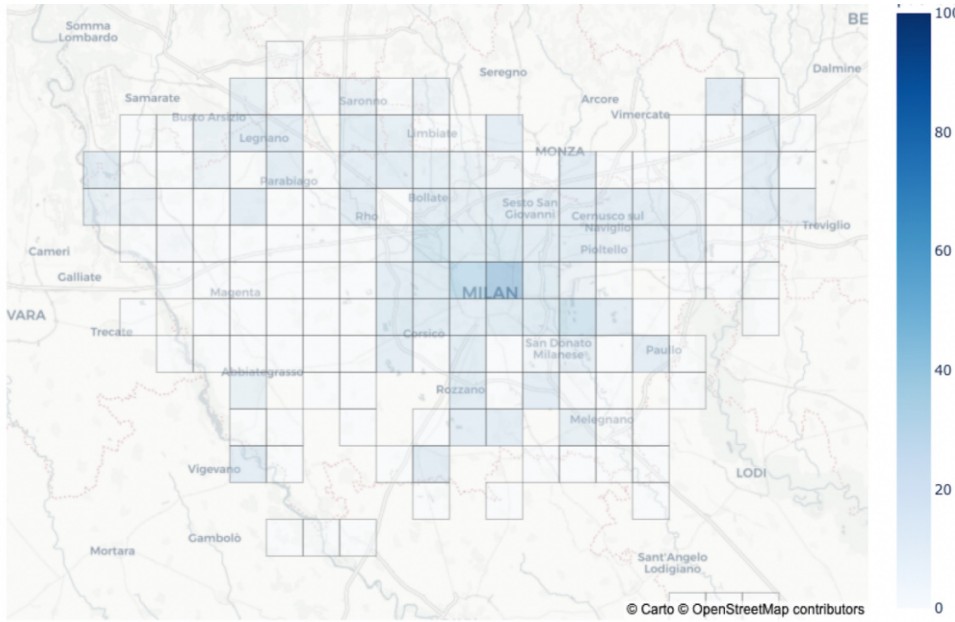

**Figure 6.** Public Transportation Accessibility Index (Environment Pillar), 2021: Province of Milan.

*3.2. Pursuing Opportunities: Applying the TSI to Investigate Potential Implications on Different Tourism-Related Areas*

The second round of tests explored potential uses of the TSI in identifying practical implications and contributions for all users, including destination stakeholders.

3.2.1. Destination Analysis

A first point was related to analyzing a single destination, in particular assessing sustainability at a sub-destination level. Within a destination, tourism activity may be distributed unequally, being highly concentrated in some subareas and less concentrated in other subareas. Because the Bing Tile System is essentially a hierarchical mosaic of the Earth's surface in its Mercator representation, it is possible to change the scale to exhibit a different fine-grained geographical grouping. The size of the tile halves each time the level increases, starting with level 0 that covers the entire Mercator projection of the globe. We chose a level of detail with a Bing Tile size of 16 corresponding to a width of 611.5 × 611.5 m, which enabled a more thorough analysis of sustainability at the sub-destination level. Figure 7 shows an application on the municipalities of Milan and Florence of one of the TSI subcomponents, tourism pressure, i.e., the relationship between visitors and residents: the lighter the tile, the higher the pressure. The data visualization in Figure 7 shows how remarkably the different neighborhoods can diverge and the consequent significance of a sub-destination level measurement, in terms of managerial implications.

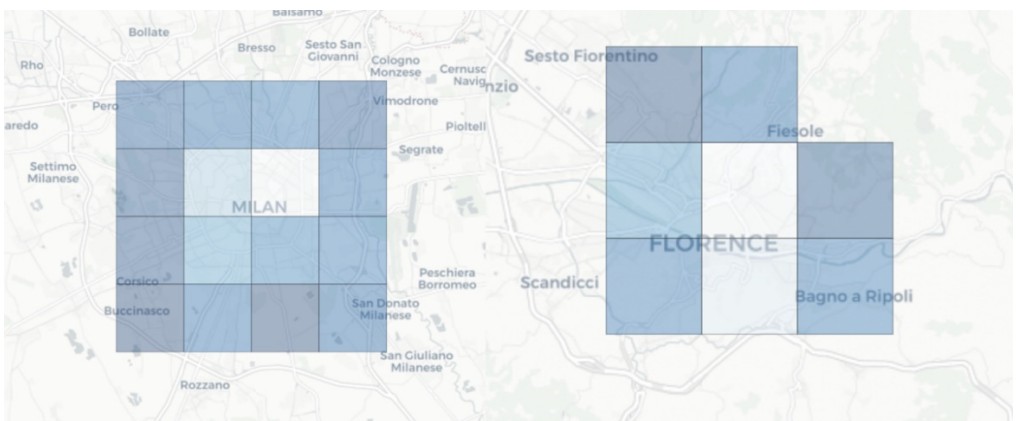

**Figure 7.** Tourism Pressure: Municipalities of Milan and Florence.

Furthermore, the monthly updates permit a real-time analysis of the evolution of the sustainability assessment, both for the destination as a whole and for sub-destination levels. Figure 8 shows the application of the overtourism pillar to the municipality of Florence in the same month over the last 3 years. the lighter the tile, the higher the overtourism level.

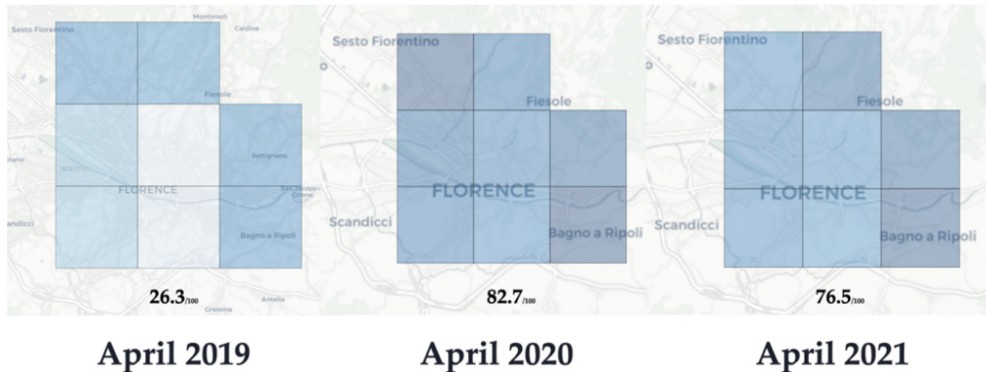

**Figure 8.** Overtourism pillar, 2019 to 2021: Municipality of Florence.

It is possible to use the TSI and its subcomponents as a tool to monitor, on a monthly basis, several dimensions of the tourism phenomenon, even those that are difficult to track with official statistics. Figure 9 shows the application of sub-indicator monitoring of short-term rentals in the province of Milan during 3 months in 2021. The lighter the tile, the higher the activity. It is evident that, as the pandemic restrictions were easing, the peer-to-peer accommodation sharing market began to expand its boundaries and impacts.

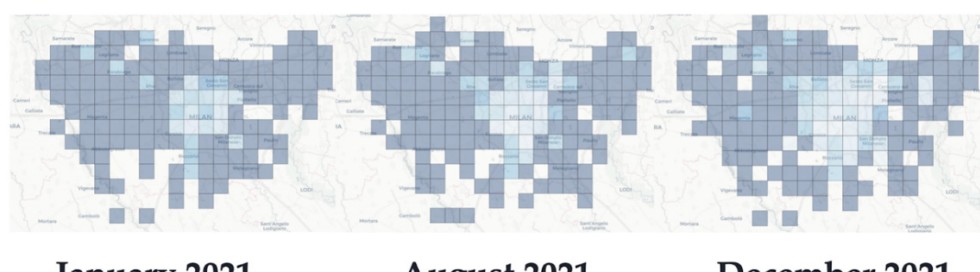

**Figure 9.** Activity of short-term rentals: Province of Milan.

### 3.2.2. Destinations Comparison

Comparisons of the sustainability assessments of different types of tourism destinations with monthly updates have not yet been made. Our methodology enables the comparison of the TSI indicators on a holistic basis, as well as a comparison of the TSI's subcomponents, to facilitate the benchmarking of different dimensions of tourism sustainability. Figure 10 shows the application of the four TSI pillars in a comparison of tourism in Florence and Venice. Both cities have a similar quadrilateral shape in Figure 10's spider plot. The historical city of Venice has a higher grade of sustainability on the environmental and social-cultural aspects, while Florence registers a higher level of sustainability on the overtourism and destination pillars.

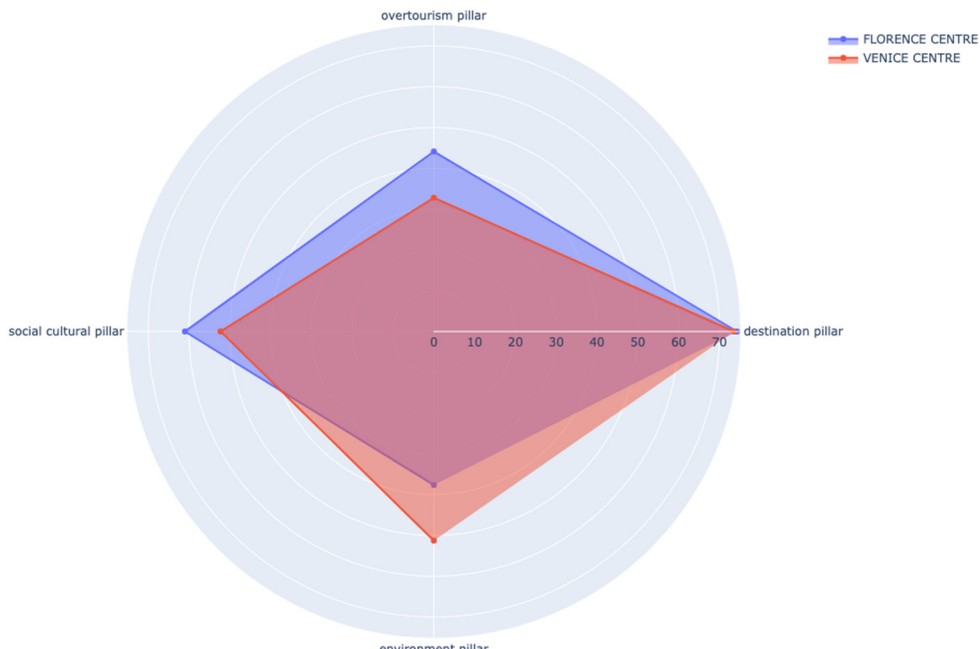

**Figure 10.** TSI pillars: Venice vs. Florence.

A second comparison was applied to the two places that are holding the Winter Olympic Games in 2026, Milan and Cortina. The quadrilateral shapes for the two destinations in Figure 11's spider plot are quite different. From a managerial perspective, Cortina and Milan seem to compensate each other in terms of sustainability. This could

be interpreted as a new way to assess the model of joint tourism projects involving different destinations.

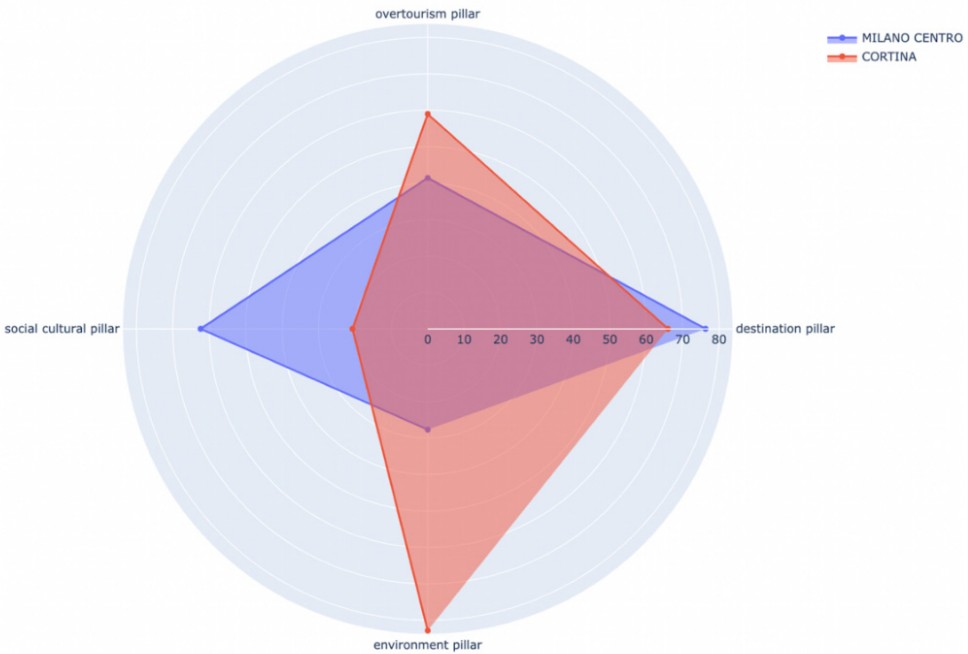

**Figure 11.** TSI pillars: Milan vs. Cortina.

## 4. Discussion and Conclusions

In this paper, a pragmatic and original approach to establishing a Tourism Sustainability Index (TSI) was presented. With the ETIS model as a reference, and bearing in mind the challenges of the ETIS applicability, we proposed a combination of open data sources and proprietary data, based on sentiment analysis, to achieve a scalable index, updated monthly. The core of this index was its capacity for estimating both objective and subjective parameters that can affect the sustainability of a destination. While the objective parameters were deduced by analysis of a set of open data sources, the subjective parameters were mostly based on sentiment analysis that was derived from UGCs that were related to a certain number of POIs within any type of tourism destination. Spatially, the index may be computed at both the destination level (i.e., with respect to the portion of territory that is of interest to tourists) and the tile level, using the Bing Maps Tile System. This type of analysis may be useful for monitoring, comparing, and planning sustainability policies for implementation by destination management organizations (DMOs) and private businesses.

### 4.1. Theoretical Contribution

The theoretical contributions of this paper are greater than the studies of the different subject domains. From a tourism perspective, the first contribution was applying sentiment analysis via a rigorous and scientific process, which was a proficient way to determine user satisfaction as an essential topic when measuring tourism sustainability, at both the business level and the destination level. Moreover, using sentiment analysis with a wide data lake of heterogeneous UGCs, as proposed and tested in the paper, boosted the need for a reconsideration of previous approaches in measuring sustainability, from analytical and synthetic points of view. Therefore, the aggregate method of the ETIS, as described in the relevant literature, was juxtaposed with a synthetical index, the TSI, whose components and subcomponents are themselves indices that are reconciled and computed on weighted averages, rather than via summations of absolute values. Our proposed change in paradigm led to a more holistic way of addressing the measurement of tourism sustainability, as a process involving not only different dimensions (social-cultural, economic, environmental, and governance) but also a consideration of how these dimensions interact with each other,

mostly in not-linear ways, within the common factor of the user satisfaction. Consequently, from a data-science point of view, the main contribution of this paper is providing a consistent approach to relating data sources with different characteristics, spatial and temporal granularities, and updating frequencies, in order to build a single representative model for tourism sustainability with a synthetic index. As each element was validated and reconciled with all of the other TSI subcomponents, the normalization of results without natural references was derived by taking into consideration significant percentiles of the values as maxima and minima. From a strictly engineering point of view, the adoption of a data ingestion technique that allows a continuous updating of the values is another staple we wanted to underline as fundamental when computing data with high variance for a common characteristic (i.e., frequency). The ingestion system independently evaluates each external data source, seeks any changes, and preprocesses them for direct use in meeting the researchers' purposes. The theoretical and technical effort required for the ingestion process, which involves a quite heterogeneous team in terms of competencies and skills—database specialists, software engineers, infrastructure engineers, data engineers, domain experts— is not often easy to reproduce in a classical university context. Companies considering such an effort a part of their core business may provide great assistance in supporting researchers with the process, and partnerships may be promoted and created. Finally, involving different subject domains leads to another point and yet another contribution: i.e., the research process must consider including a data visualization tool so that each member of the team is aware of the progress and the results. For projects regarding tourist destinations, we propose using a geographical map as the main reference to provide immediately an intuitive indication of the data, while the detail of each subcomponent can be considered by way of a separate panel. In addition, an easily readable and shareable data visualization tool helps greatly in reporting and presenting results to a wider audience, thereby increasing the impact and dissemination of the research.

### 4.2. Managerial Implications

The first round of tests applying TSI to a heterogenous area, such as the Province of Milan, led to a finding of a direct correspondence with other data sources, in line with was reported or evaluated by the other entities and research centers mentioned. The second round of tests explored potential uses of the TSI and identified several practical implications and contributions. The managerial implications we outlined for destination management organizations (DMOs), tourism authorities, and private business managers, were exemplified by data visualization via the application of the TSI for several tourism-related areas. Destination analysis, monitoring, and comparison led all users, including destination stakeholders, to potentially consider the index as a fundamental management tool that is updated monthly, highlighting the importance of widespread awareness in including sustainability in strategic planning and operative actions. Assessing the level of sustainability for any tourism destination, worldwide, is readily achievable without any data from the destinations themselves; this was one of the main challenges created by a survey-only-based approach. Computing the TSI and its subcomponents on a Bing Tile System enables the assessment of sustainability for any area, even for destinations whose boundaries do not necessarily follow administrative borders. In fact, the literature has pointed out that administrative boundaries, or other political boundaries, may potentially divide destinations, in particular with respect to natural areas, and potentially hinder tourism development [51]. Consumers' perspectives should be taken into consideration, even if actual attempts of structuring tourism geographies based on visitors' consumption patterns are scant [52].

### 4.3. Limitations and Future Research

The ongoing validation process, consisting of reviews by domain experts of the methodology and the possible managerial implication, will be followed by an initial implementation in 2022. A limitation of this process is that expenditure data patterns are

not included in the actual MVP assessment of the TSI, nor are labor-related statistics. We followed paths to extend the effectiveness and the context of sustainability analysis by integrating new data sources and refining the semantic analysis tools that were used to create the TSI. The opportunity to use proprietary data as the backbone of the TSI, which is based on data sources that are self-updating, may also be a limitation. In fact, the use of proprietary data does not allow anyone to immediately reproduce the results presented in this paper, which can be achieved only by organizations that are able to apply the process and techniques described in the methodology section. Moreover, we emphasize that in this stage, the TSI and its sub-indicators were designed to focus on correlations of the different factors, but no causality analysis was performed. Investigating the causality and the statistical independence of the components may be a valuable project future research. Any collaboration with other data providers, institutions, universities, and researchers would amplify the chances of extending the scope and assessment of the TSI. Other possible future research may include new algorithms to compute sentiment analysis and/or the effectiveness of other models in assessing a destination's reputation. In fact, sentiment analysis is only one of the NLP tools that aim to detect and derive subjective perception. Other tools (e.g., emotion analysis) could be integrated into future implementations of the TSI in order to further assess users' feelings, beyond satisfaction. From an operational point of view, the TSI allows the projection of each dimension to be analyzed on the map, for the purpose of evaluating its geographical distribution and the mutual relations between different areas. Future research may start from a quantitative study of these relationships with the aim of building a forecasting model with respect to the carrying capacity of different destinations. A more experimental future research project might focus on modelling benchmark analysis for different destinations or tiles, based on data-driven clusters of tourism destinations, to identify any common patterns that are beyond the actual product-oriented classification. Finally, the TSI approach, which is based on data sources that are self-updating, may be extended to a wider context outside the tourism sector. Sustainable development goals (SDGs) have a general context within which a whole society may be evaluated. From this point of view, the TSI may lead to definitions of additional general but effective sub-indicators that could provide data-driven suggestions for public and private decision makers.

**Author Contributions:** Conceptualization, D.D.M. and R.B.; methodology, D.D.M. and R.B.; validation, D.D.M., R.B. and L.D.S.; formal analysis, D.D.M., R.B. and L.D.S.; data visualization, L.D.S.; resources, D.D.M., R.B. and L.D.S.; writing—original draft preparation, D.D.M., R.B. and L.D.S.; writing—review and editing, D.D.M. All authors have read and agreed to the published version of the manuscript.

**Funding:** This research received no external funding.

**Institutional Review Board Statement:** Not applicable.

**Informed Consent Statement:** Not applicable.

**Data Availability Statement:** Data sharing is not available, due to the presence of proprietary data.

**Acknowledgments:** This study was supported by The Data Appeal Company.

**Conflicts of Interest:** The authors declare no conflict of interest.

## Abbreviations

| | |
|---|---|
| DMO | Destination Management Organization |
| ECMWF | European Centre for Medium-Range Weather Forecasts |
| ESA | European Space Agency |
| ETIS | European Tourism Indicators System |
| EUMETSAT | European Organization for the Exploitation of Meteorological Satellites |
| MICE | Meetings, Incentives, Conferences, and Exhibitions |

| MVP | Minimum Viable Product |
|---|---|
| NER | Name Entity Recognition |
| NLP | Natural Language Processing |
| OTA | Online Travel Agency |
| POI | Point of Interest |
| TDAC | The Data Appeal Company |
| TSI | Tourism Sustainability Index |
| UGC | User-Generated Content |
| UN | United Nations |
| UNWTO | World Tourism Organization |

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
