# Peer review of "Tourism Sustainability Index: Measuring Tourism Sustainability Based on the ETIS Toolkit, by Exploring Tourist Satisfaction via Sentiment Analysis"

_sustainability, doi:10.3390/su14138049_

Round 1

Reviewer 1 Report

The study entitled Tourism Sustainability Index: Measuring tourism sustainability based on the ETIS toolkit, by exploiting tourist satisfaction via Sentiment Analysis presents an interesting topic and an original perspective on it also matching the theme of the Sustainability journal Special Issue entitled “Sustainable Tourism and Tourist Satisfaction”.

However the research paper has several major gaps as explained below:

-          The paper has a clear and logical introduction but fails to explain its contribution from both methological and aimed results point of view to the existing knowledge on the topic. The aim, objectives, research hypotheses of the study are not convincingly expressed.

-          The paper does not clearly and convincingly explain the relation between the ETIS toolkit and tourist satisfaction. Moreover there seem to exist a contradiction in the discourse of the paper declaring at Lines 66-68 that „Despite the challenges, ETIS is still the most known, cited, and valuable methodology to implement a quantitative and effective measurement of tourism sustainability at destinations’ level” while the research attempt is to introduce tourist satisfaction perceived rather as a qualitative variable in the analysis.

-          The methodology contains rather general explanations on data sources and automatization process and on ETIS methodology. There are no convincing explanations on how each variable of the ETIS methodology was selected/eliminated from the analysis and there is lack of coherence and contradictions among different methodological parts. For example in Table 1 C4 Inclusion accessibility index is marked as having no connection with TSI and sentiment analysis approach whereas below at point 2.3.3 at lines 307-309 the authors explain “A special focus has been given also to accessibility, seen as how many POIs are accessible to disabled, but also identifying how the whole destination is accessible in terms of public transportation” emphasizing the importance of both C4 and D1 for the study.

-          There is a general lack of coherence among title, introduction, methodology and results. According to the title the main methodological focus should be on tourist satisfaction  through Sentiment Analysis in relation or as part of TSI. Further in the article it is not clearly emphasized how this concept of satisfaction is integrated in the analysis. It is also not clear if the authors decided to follow the index list in table 1 or the pillars approach and which are the variables remaining and why. Is the lack of data the elimination criterion ? And if so this endangers the results overall. Are there other reasons for indicators selection as connected to this very topic ?

-          Data sources are too generally and not properly explained. Exact references for consulted studies, what data / indexes were gathered, for what period, from which institution should be stated and would certify the scientific discourse of the paper. Both Table 1 and text should be completed in this respect.

-       It is no clear how data offered by different institutions matched at the studied territorial scale and on which criteria the selection of the study case was done. These aspects are superficially treated in the first part of the results and should be motivated in the introduction and also in the methodological part.

-          Taken into consideration all the above comments the results are not convincing (e.g. values of the pillars in figure 1) and need to be better motivated by explanations which should be related to all the other parts of the paper. Moreover certain results also contradict methodology (e.g. Figure 2 focusing on Tourism pressure whereas Table 1 states that both Tourism pressure and Tourism Supply Pressure are not connected with sentiment analysis).

-          Results and discussions do not contain references on tourist satisfaction and sentiment analysis which is supposed to be part of methodology and of the main focus of the article

-          The major gaps on methodology and results and the overall incoherence of the article need to be addressed to improve the paper.

Author Response

Dear reviewer,

Thank you very much for your precious comments. We applied major revisions to the manuscript according to all the reviews we received. In detail:

  •          The paper has a clear and logical introduction but fails to explain its contribution from both methological and aimed results point of view to the existing knowledge on the topic. The aim, objectives, research hypotheses of the study are not convincingly expressed.

Response: We underlined in a more explicit way that the main contribution of this paper lies in presenting an Index to measure sustainability based on the ETIS toolkit, by exploiting tourist satisfaction via Sentiment Analysis and further combining it with open data sources. The research hypothesis is that the Sentiment Analysis that can be used non only as a scientific process to define and highlight the tourist satisfaction as a key point to be considered for measuring sustainability, but also to infer the Index, which is consequently based on data sources which are self-updating and at the same time maintaining an international comparability and an applicability to any typology of tourism destination. Moreover, a pragmatic and original approach to the definition of a Tourism Sustainability Index (TSI) is presented: a geo-referenced indicator maintaining an international comparability and an applicability to any typology of tourism destination, even at sub-destination level, which leads to be an useful tool for any stakeholders at destination level for the implementation of an actual measurement of tourism sustainability. 

  •          The paper does not clearly and convincingly explain the relation between the ETIS toolkit and tourist satisfaction. Moreover there seem to exist a contradiction in the discourse of the paper declaring at Lines 66-68 that „Despite the challenges, ETIS is still the most known, cited, and valuable methodology to implement a quantitative and effective measurement of tourism sustainability at destinations’ level” while the research attempt is to introduce tourist satisfaction perceived rather as a qualitative variable in the analysis.

Response: We added an explanation on the role of tourist satisfaction in carrying capacity and secondly in the ETIS toolkit, as an important dimension to estimate tourism sustainability. Since tourism satisfaction is relevant to assess sustainability, this explains one of the reasons for the TSI index to be based on the Sentiment Analysis, acting as a complement (not substitute) of the ETIS Toolkit, to solve some of the challenges arising in actual application. 

  •          The methodology contains rather general explanations on data sources and automatization process and on ETIS methodology. There are no convincing explanations on how each variable of the ETIS methodology was selected/eliminated from the analysis and there is lack of coherence and contradictions among different methodological parts. For example in Table 1 C4 Inclusion accessibility index is marked as having no connection with TSI and sentiment analysis approach whereas below at point 2.3.3 at lines 307-309 the authors explain “A special focus has been given also to accessibility, seen as how many POIs are accessible to disabled, but also identifying how the whole destination is accessible in terms of public transportation” emphasizing the importance of both C4 and D1 for the study.

Response: We added details on the links between the ETIS criteria and the TSI sub-indicators both in the methodology and results sections, detailing also the main criteria of selection.

  •          There is a general lack of coherence among title, introduction, methodology and results. According to the title the main methodological focus should be on tourist satisfaction  through Sentiment Analysis in relation or as part of TSI. Further in the article it is not clearly emphasized how this concept of satisfaction is integrated in the analysis. It is also not clear if the authors decided to follow the index list in table 1 or the pillars approach and which are the variables remaining and why. Is the lack of data the elimination criterion ? And if so this endangers the results overall. Are there other reasons for indicators selection as connected to this very topic ?

Response: We emphasized the role of tourism satisfaction via Sentiment Analysis in each section of the paper and better explained how we reached the actual proposed TSI, pillars and sub-components. 

  •          Data sources are too generally and not properly explained. Exact references for consulted studies, what data / indexes were gathered, for what period, from which institution should be stated and would certify the scientific discourse of the paper. Both Table 1 and text should be completed in this respect.

Response: We better explained how we reached the actual proposed TSI in terms of the sources, indicators selection and how the different data sources were combined at spatial, temporal scale.

  •      It is no clear how data offered by different institutions matched at the studied territorial scale and on which criteria the selection of the study case was done. These aspects are superficially treated in the first part of the results and should be motivated in the introduction and also in the methodological part.

Response: We better explained how we reached the actual proposed TSI in terms of the sources, indicators selection and how the different data sources were combined at spatial, temporal scale.

  •          Taken into consideration all the above comments the results are not convincing (e.g. values of the pillars in figure 1) and need to be better motivated by explanations which should be related to all the other parts of the paper. Moreover certain results also contradict methodology (e.g. Figure 2 focusing on Tourism pressure whereas Table 1 states that both Tourism pressure and Tourism Supply Pressure are not connected with sentiment analysis).

Response: We revised the results / discussion section according to the different reviews, adding also more results related to the sentiment analysis.

  •          Results and discussions do not contain references on tourist satisfaction and sentiment analysis which is supposed to be part of methodology and of the main focus of the article

Response: We revised the results / discussion section according to the different reviews, adding also more results related to the sentiment analysis.

  •          The major gaps on methodology and results and the overall incoherence of the article need to be addressed to improve the paper.

Response: We applied major revisions to all the paper accordingly, to reduce the perceived incoherence.

Kind regards,

The authors

Reviewer 2 Report

The topic is highly interesting for the current situation in tourism industry. However, the author should explain more clearly how the areas of study are suitable for the study. 

The literature was adequate. 

For the methodology, the author should explain how the selected method is very appropriate for the current study, meaning that the author should highlight the important features of the methodology. 

The results of the study were well-presented with figures. This is helpful to explain the important findings. However, ,the discussion with the past research was still too limited. The author should provide more details about the discussion. 

The conclusions were too short. The author should provide greater details about the theoretical contribution, and practical contributions for the related stakeholders. In addition, directions for future research should also be addressed and limitations of the study should also be mentioned.  

Author Response

Dear reviewer,

Thank you very much for your precious comments. We applied major revisions to the manuscript according to all the reviews we received. In detail:

For the methodology, the author should explain how the selected method is very appropriate for the current study, meaning that the author should highlight the important features of the methodology. 

Response: We revised the methodology to better explain how we reached the actual proposed TSI in terms of the sources, indicators selection and how the different data sources were combined at spatial, temporal scale. We also underlined in a more explicit way that the main contribution of this paper lies in presenting an Index to measure sustainability based on the ETIS toolkit, by exploiting tourist satisfaction via Sentiment Analysis and further combining it with open data sources. The research hypothesis is that the Sentiment Analysis that can be used non only as a scientific process to define and highlight the tourist satisfaction as a key point to be considered for measuring sustainability, but also to infer the Index, which is consequently based on data sources which are self-updating and at the same time maintaining an international comparability and an applicability to any typology of tourism destination. Moreover, a pragmatic and original approach to the definition of a Tourism Sustainability Index (TSI) is presented: a geo-referenced indicator maintaining an international comparability and an applicability to any typology of tourism destination, even at sub-destination level, which leads to be an useful tool for any stakeholders at destination level for the implementation of an actual measurement of tourism sustainability. 

The results of the study were well-presented with figures. This is helpful to explain the important findings. However, ,the discussion with the past research was still too limited. The author should provide more details about the discussion. 

Response: We deeply revised the results / discussion / conclusions sections according to the reviews, adding also more details on the past research, more results related to the sentiment analysis, trying to extend the practical contribution for the related stakeholders. We underlined the directions for future research and implementation.

The conclusions were too short. The author should provide greater details about the theoretical contribution, and practical contributions for the related stakeholders. In addition, directions for future research should also be addressed and limitations of the study should also be mentioned. 

Response: We deeply revised the results / discussion / conclusions sections according to the reviews, adding also more details on the past research, more results related to the sentiment analysis, trying to extend the practical contribution for the related stakeholders. We underlined the directions for future research and implementation.

Kind regards,

The authors

Reviewer 3 Report

The paper is well written and easy to read. The authors did a fine job. However, I have some suggestions before recommending it for publication:

Line 233 - you wrote "A tile size of 16..." 16 of what exactly? We need some measures.

The section 2.2.1. Open data is too widely described. The data sources would suffice. Better state the Copernicus and WB data products.

The results should be better described. Currently, the results describe more what the results should be or should show instead of the actual research results. Many of the results are shown in the discussion section. Please move them to the results section.

Author Response

Dear reviewer,

Thank you very much for your precious comments. We applied major revisions to the manuscript according to all the reviews we received. In detail:

Line 233 - you wrote "A tile size of 16..." 16 of what exactly? We need some measures.

Response: We added details on tile size.

The section 2.2.1. Open data is too widely described. The data sources would suffice. Better state the Copernicus and WB data products.

Response: We reduced this section, maintaining a certain grade of detail due to our aim to be useful for newbies using the data sources described.

The results should be better described. Currently, the results describe more what the results should be or should show instead of the actual research results. Many of the results are shown in the discussion section. Please move them to the results section.

Response: We deeply revised the results / discussion / conclusions sections according to the reviews, adding also more details on the past research, more results related to the sentiment analysis, trying to extend the practical contribution for the related stakeholders. We underlined the directions for future research and implementation.

Kind regards,

The authors

Reviewer 4 Report

The aim of this paper is to propose a new indicator (TSI) to measure the sustainability of tourism destinations. The idea is interesting. However, the paper needs some further improvement. The authors are invited to revise the manuscript to address the following points:

·       The introduction is too long. It is suggested that the authors write a new introduction following the ‘Instructions for Authors’ and rename this section into a theoretical background.

·       In section 1.1, the authors could provide more information about the ETIS model. For example, what exactly it is (e.g. a management tool, a monitoring system, an information tool), the benefits for the destinations that will use it, etc.

·       The authors argue that the ETIS implementation faces many challenges and obstacles, especially at subnational level. The authors rely mainly on this argument to support the need to develop a new model that will address these obstacles. However, the European Commission itself, which developed the ETIS system, claims that it has already successfully implemented it in more than 100 destinations, both nationally and at regional / city level (https://ec.europa.eu/growth/sectors/tourism/offer/sustainable/indicators_en). In fact, one of the case studies EC provides concerns Milan, which was also used by the authors to implement their own model. Therefore, the authors should support with more arguments the weaknesses that they consider the ETIS model has. Otherwise, it would be better for them to present their own model as an alternative methodology for measuring the sustainability of destinations that may complement the ETIS model or used instead, rather than as a model that solves the problems faced by the ETIS model.

In the methodology section, the authors should indicate when the data were collected, when the analysis was performed, etc. (time period).

Author Response

Dear reviewer,

Thank you very much for your precious comments. We applied major revisions to the manuscript according to all the reviews we received. In detail:

  • The introduction is too long. It is suggested that the authors write a new introduction following the ‘Instructions for Authors’ and rename this section into a theoretical background.

Response: Since other reviewers asked to underline in a more explicit way the main contribution of this paper, the research hypothesis and the approach, it was not easy to reduce it. We revised this section according to all the reviewers points of view.

  •     In section 1.1, the authors could provide more information about the ETIS model. For example, what exactly it is (e.g. a management tool, a monitoring system, an information tool), the benefits for the destinations that will use it, etc.

Response: We briefly added more information about the ETIS model, the benefits for users and the role of tourist satisfaction.

  •     The authors argue that the ETIS implementation faces many challenges and obstacles, especially at subnational level. The authors rely mainly on this argument to support the need to develop a new model that will address these obstacles. However, the European Commission itself, which developed the ETIS system, claims that it has already successfully implemented it in more than 100 destinations, both nationally and at regional / city level (https://ec.europa.eu/growth/sectors/tourism/offer/sustainable/indicators_en). In fact, one of the case studies EC provides concerns Milan, which was also used by the authors to implement their own model. Therefore, the authors should support with more arguments the weaknesses that they consider the ETIS model has. Otherwise, it would be better for them to present their own model as an alternative methodology for measuring the sustainability of destinations that may complement the ETIS model or used instead, rather than as a model that solves the problems faced by the ETIS model.

Response: We underlined in a more explicit way that the main contribution of this paper lies in presenting an Index to measure sustainability based on the ETIS toolkit, by exploiting tourist satisfaction via Sentiment Analysis and further combining it with open data sources. The research hypothesis is that the Sentiment Analysis that can be used non only as a scientific process to define and highlight the tourist satisfaction as a key point to be considered for measuring sustainability, but also to infer the Index, which is consequently based on data sources which are self-updating and at the same time maintaining an international comparability and an applicability to any typology of tourism destination. Moreover, a pragmatic and original approach to the definition of a Tourism Sustainability Index (TSI) is presented: a geo-referenced indicator maintaining an international comparability and an applicability to any typology of tourism destination, even at sub-destination level, which leads to be an useful tool for any stakeholders at destination level for the implementation of an actual measurement of tourism sustainability. We added also an explanation on the role of tourist satisfaction in carrying capacity and secondly in the ETIS toolkit, as an important dimension to estimate tourism sustainability. Since tourism satisfaction is relevant to assess sustainability, this explains one of the reasons for the TSI index to be based on the Sentiment Analysis, acting as a complement (not substitute) of the ETIS Toolkit, to solve some of the challenges arising in actual application.

In the methodology section, the authors should indicate when the data were collected, when the analysis was performed, etc. (time period).

Response: We added details on the links between the ETIS criteria and the TSI sub-indicators both in the methodology and results sections. We better explained how we reached the actual proposed TSI in terms of the sources, indicators selection and how the different data sources were combined at spatial, temporal scale as well as more details on when data were collected and analysis performed.

Kind regards,

The authors

Round 2

Reviewer 1 Report

The new version of the study entitled Tourism Sustainability Index: Measuring tourism sustainability based on the ETIS toolkit, by exploiting tourist satisfaction via Sentiment Analysis presents supplementary arguments which support the scientific discourse of the paper.

The formulation of research hypothesis was an important adding which presents in more explicit terms the logic of the paper.

The availability of data is an important aspect explained by the authors and the logic of eliminating data because not available is completely understood. However more rigurous explanations are needed to answer how and why these eliminations didn’t endanger the scientific results of the paper.

Another question mark that remains somehow unsolved for this paper is the one concearning data sources. Proprietary data might be accepted as declared terms in an experimental empirical scientific discourse but lowers the credibility of the article overall.

Results are now more commented but because of missing clarifications on data and on their connectivity some of the explanations seem rather superficial and are not consistently scientifically argumented. Otherwise the mapping of data and indicators do not necessarily reveal the causality of factors and revealing more in depth sequences of the analysis or complementary views might be necessary  (e.g. statistic tests).

The article proposes an interesting approach in terms of cross interfering Sentiment Analysis to TSI and might inspire other researchers in improving the method.

I recommend the authors to clarify some of the aspects underlined above for the publication of their paper.

Author Response

Dear reviewer,

Thank you very much for your precious comments. We applied revisions to the manuscript according to your reviews, in detail:

  • The availability of data is an important aspect explained by the authors and the logic of eliminating data because not available is completely understood. However more rigorous explanations are needed to answer how and why these eliminations didn’t endanger the scientific results of the paper.

We added more explanations on the author's selection and how it didn’t endanger the results maintaining a good representation of the main dimensions and aspects of the ETIS pillars.

  • Another question mark that remains somehow unsolved for this paper is the one concerning data sources. Proprietary data might be accepted as declared terms in an experimental empirical scientific discourse but lowers the credibility of the article overall.

Using proprietary data is both an opportunity and a limitation. The opportunity lies in being able to create and test a TSI based on data sources which are self-updating for any destination. The main limitation is that using proprietary data, the possibility for anyone for replicating the test could be somehow affected. We added both considerations as well as some figures on proprietary data to validate the credibility of the article overall.

  • Results are now more commented but because of missing clarifications on data and on their connectivity some of the explanations seem rather superficial and are not consistently scientifically argumented. Otherwise the mapping of data and indicators do not necessarily reveal the causality of factors and revealing more in depth sequences of the analysis or complementary views might be necessary  (e.g. statistical tests).

In this stage we focused on correlation of the factors, not on the causality. We added it as a limitation of our research, as well as potential future research. 

  • The article proposes an interesting approach in terms of cross interfering Sentiment Analysis to TSI and might inspire other researchers in improving the method.

We added this comment as another potential future research. 

  • I recommend the authors to clarify some of the aspects underlined above for the publication of their paper.

We did our best to clarify them all.

Kind regards,

The authors

This manuscript is a resubmission of an earlier submission. The following is a list of the peer review reports and author responses from that submission.